# Pathophysiological and Molecular Basis of the Side Effects of Ticagrelor: Lessons from a Case Report

**DOI:** 10.3390/ijms241310844

**Published:** 2023-06-29

**Authors:** Daniel Cesarini, Iacopo Muraca, Martina Berteotti, Anna Maria Gori, Andrea Sorrentino, Alessia Bertelli, Rossella Marcucci, Renato Valenti

**Affiliations:** 1S.O.D. Cardiologia Interventistica d’Urgenza, AOU Careggi, 50134 Florence, Italy; daniel.cesarini@unifi.it (D.C.); iacopo.muraca@unifi.it (I.M.); valentir@aou-careggi.toscana.it (R.V.); 2Atherothrombotic Center, Department of Experimental and Clinical Medicine, University of Florence, AOU Careggi, 50134 Florence, Italy; annamaria.gori@unifi.it (A.M.G.); andrea.sorrentino@unifi.it (A.S.); alessia.bertelli@unifi.it (A.B.); rossella.marcucci@unifi.it (R.M.)

**Keywords:** ticagrelor, dyspnea, bradyarrhythmias, pacemaker, adenosine, myocardial infarction

## Abstract

Ticagrelor is currently considered a first-line choice in dual antiplatelet therapy (DAPT) following revascularization of acute coronary syndrome (ACS). However, its use is correlated with an increased incidence of two side effects, dyspnea and bradyarrhythmias, whose molecular mechanisms have not yet been defined with certainty and, consequently, neither of the therapeutic decisions they imply. We report the case of a patient with acute myocardial infarction treated with ticagrelor and aspirin as oral antithrombotic therapy after primary percutaneous coronary intervention (PCI), manifesting in a significant bradyarrhythmic episode that required a switch of antiplatelet therapy. Starting from this case report, this article aims to gather the currently available evidence regarding the molecular mechanisms underlying these side effects and propose possible decision-making algorithms regarding their management in clinical practice.

## 1. Introduction

Ticagrelor is an oral antiplatelet agent and reversible and direct inhibitor of the adenosine diphosphate P2Y12 receptor. In the PLATO trial, it demonstrated greater effectiveness over clopidogrel when used in addition to aspirin in reducing rates of ischemic complications and stent thromboses [1], so it is considered a first-line drug in antiplatelet therapy after acute coronary syndrome (ACS) revascularization (Class I, Level B ESC guidelines; Class I LOE B-R ACC/AHA guidelines) [2,3]. However, its use has been associated with three main side effects: bleeding, dyspnea, and bradyarrhythmias (sinus bradycardia, sinoatrial blocks, and atrioventricular (AV) blocks). These side effects may lead to a switch or a de-escalation of antiplatelet therapy to clopidogrel, with the potential drawback of the reduced antithrombotic efficacy of double antiplatelet therapy (DAPT). While bleeding events have been extensively addressed, the pathogenetic mechanism of dyspnea and bradyarrhythmias has not yet been fully elucidated.

Starting from a real case presentation, the purpose of this report is therefore to gather the currently available scientific evidence, focusing on the pathogenesis and clinical impact of these side effects. The incidence and clinical impact of bleeding will not be the focus of this document.

## 2. Case Description

A 57-year-old male patient with multiple cardiovascular risk factors, including hypertension, dyslipidemia, and a family history of coronary artery disease (CAD), presented to the emergency department with chest pain lasting over 12 h. Despite previous episodes of chest pain and dyspnea, his electrocardiogram (ECG) showed normal sinus rhythm with QS complexes in the inferior leads. High-sensitive serum troponin T (hs-TnT) levels did not indicate a significant delta in the serial blood tests. Echocardiography revealed akinesia of the basal-mid segment of the inferior wall and mildly decreased systolic function.

The patient was transferred to the cath lab where coronary angiography revealed occlusion due to massive thrombosis in the right coronary artery and stenosis in the left anterior descending artery. Percutaneous coronary intervention (PCI) was performed, successfully restoring blood flow in the right coronary artery and resolving the patient’s symptoms. Medication administered during the procedure included unfractionated heparin, a cangrelor bolus plus continuous infusion, a ticagrelor bolus, and aspirin.

On the first day after the procedure, the patient experienced a brief loss of consciousness. With continuous ECG monitoring, there was evidence of a sinoatrial block of more than 15 s followed by two more pauses of about 4 s each (Figure 1).

Repeated echocardiography and coronary angiography showed no new abnormalities or periprocedural complications. Based on the duration of the pauses and the clinical presentation, pacemaker implantation was indicated according to current ESC guidelines [4]. However, considering the possibility of ticagrelor-induced sinoatrial block, the patient was switched to prasugrel and closely monitored without undergoing pacemaker implantation.

Further investigations, including carotid sinus massage and an electrophysiological study, did not reveal any significant abnormalities (AH interval 95 ms, HV interval 40 ms, and correct sinus node recovery time: 220 ms). A loop recorder was implanted, which recorded no further episodes of sino-atrial block. Notably, blood samples taken at 24 h showed significantly higher adenosine levels (C: 104.3 ng/mL) compared to both a patient on clopidogrel admitted for ACS (B: 36.82 ng/mL) and a patient not on antiplatelet therapy (A: 21.28 ng/mL) (Figure 2).

In the following weeks, the patient continued therapy with aspirin and prasugrel, and no other side effects or events occurred.

## 3. Discussion

### 3.1. Epidemiology and Clinical Presentation of Dyspnea and Bradyarrhythmia after Ticagrelor Intake

This case report offers the opportunity to review two of the most common side effects of ticagrelor, dyspnea, and bradyarrhythmias.

According to different studies, dyspnea is reported, ranging from 10% up to 38% [5,6]. In the PLATO trial, a dyspnea adverse event was reported by 13.8% of patients treated with ticagrelor and by 7.8% of patients treated with clopidogrel [1]. Similar percentages were described in the PEGASUS trial, where dyspnea was reported by 14.2% of patients taking ticagrelor 60 mg twice daily and by 5.5% of patients taking aspirin alone [7]. The observed dyspnea rates in clinical practice are even higher than those reported in randomized clinical trials, reaching 60% in some registries, despite mostly being in a mild form [8].

Ticagrelor-associated dyspnea usually begins roughly a few hours after starting the treatment, has a sudden onset, lasts for a minute or two, and then spontaneously vanishes. It resembles Cheyne–Stokes respiration since its intensity rises to a peak and then begins to fall and is accompanied by intense anxiety, panic, and terror. In the absence of other underlying causes of dyspnea, such as heart failure exacerbation, the patient is asymptomatic in between the episodes. Across the lungs, there are no pathological auscultatory characteristics, although severe bronchial spasms with a reduction in blood oxygen may be found. Dyspnea attacks frequently recur over the course of a few hours or even days [9].

Several studies have been carried out to determine clinical and genetic factors that predict the development of dyspnea in patients receiving ticagrelor. The data results showed that the symptom was more common in patients with two-vessel CAD (*p* = 0.034), hypothyroidism (*p* = 0.049), lower high-sensitivity ADP assessed by MULTIPLATE (ADP HS, indicative of higher platelet inhibition), and higher serum creatinine levels (*p* = 0.019) and in those who used atorvastatin lower than 80 mg (or no atorvastatin, *p* ≤ 0.001). Multivariate analysis with logistic regression analysis showed that the ADP HS value of ≤19.5 U was associated with more than a four-fold greater risk of developing dyspnea (OR = 4.07, *p* ≤ 0.001). Symptom development has also been related to certain genetic polymorphisms such as the FBG-C148T gene polymorphism or patients homozygous for the C allele (*p* = 0.014) [10].

For what causes bradyarrhythmias, in the PLATO trial, continuous ECG assessment of 3000 patients reported that 6.0% of patients in the ticagrelor cohort had ventricular pauses, which were primarily asymptomatic, sinoatrial nodal in origin, and nocturnal in nature; most of them occurred within the first week following randomization during the acute phase of ACS. However, there was no significant change in the incidence of clinically reported bradycardic episodes between ticagrelor and clopidogrel, either during the ECG evaluation period or over the subsequent 6 to 12 months of clinical follow-up. Furthermore, there was no significant change in the incidence of syncope, pacemaker implantation, cardiac arrest, or unexpected death [11].

Bradycardia and blocks in the conduction system arise approximately two hours after taking the medication, last for a short time, and then disappear; in most cases, they do not manifest clinically [9]. The most frequent type of bradyarrhythmia is sinus bradycardia (39.6%), followed by dropped beats (31.2%) and then ventricular pauses (5.8%) predominantly due to AV node pauses and secondarily due to AV blocks [11].

Considering the data provided by the distribution company in the European market [AstraZeneca S.p.A], there are currently approximately 650,000 patients receiving ticagrelor in Europe, of which around 61,000 are in Italy. According to the percentages reported in the above-mentioned randomized controlled trials (RCTs), about 8500 patients may experience dyspnea and 3700 different degrees of conduction disturbances in our country.

Given that ticagrelor administration is often concurrent with the percutaneous treatment of an ACS, bradyarrhythmic events related to myocardial infarction should be excluded. In fact, the incidence of bradyarrhythmias is not negligible in inferior STEMI, since the vascularization of the sinoatrial node and the AV node is supplied by the right coronary artery in 60% and 90% of patients, respectively. The most frequent arrhythmias related to inferior STEMI are sinus bradycardia (40% of patients in the first two hours, decreasing to 20% by the end of the first day), first-degree AV block, second-degree AV block of the Wenckebach type (Mobitz type I), and high-degree AV block. The main pathogenetic mechanism is related to ischemia of the AV node, by enhanced acetylcholine release from the infero-posterior myocardium, or possibly by making the AV node hypersensitive to the action of acetylcholine. Similar to ticagrelor arrhythmias, they are usually transient, and stable patients should undergo pacemaker implantation only if arrhythmias persist after revascularization and after exclusion of any additional potentially reversible causes (e.g., increased vagal tone, hypothyroidism, hyperkalemia, and drugs that depress conduction) [12].

### 3.2. Pathogenesis

To explain the mechanism behind these side effects, two key pathways have been identified, the first one involving a rise in extracellular adenosine levels and the second one based on the interaction between the drug and the P2Y12 receptors.

#### 3.2.1. Adenosine Hypothesis

Adenosine is a purine nucleoside produced primarily through the metabolism of ADP or adenosine triphosphate by the nucleotidases CD39 and CD73. Hypoxia, allergy stimulation, and stress all dramatically enhance the production of adenosine and its extracellular levels. Usually, cells quickly absorb extracellular adenosine in the intracellular space through sodium-independent equilibrative nucleoside transporters (ENTs) and sodium-dependent concentrative nucleoside transporters (CNTs), where it is swiftly converted to inosine or adenine nucleotides so that its half-life is only a few seconds [13,14,15]. Ticagrelor acts by blocking adenosine uptake into cells, inhibiting sodium-independent ENT 1/2, and, as a result, it lengthens the half-life and tissue concentration of extracellular nucleoside [13,16,17].

Adenosine exerts its biological effects by interacting with 4 G-protein–coupled receptors (A1, A2A, A2B, and A3): A1R and A3R are coupled to the inhibitory G protein, which inhibits adenylyl cyclase and decreases intracellular cAMP, whereas A2AR and A2BR are coupled to the stimulatory G protein which stimulates adenylyl cyclase, increasing intracellular cAMP. Of the two A2R subtypes, A2A is the high-affinity receptor and A2B is the low-affinity receptor [18,19]. This purine nucleoside stimulates vagal C fibers on the bronchial wall via A1R and A2AR, which ultimately results in dyspnea [20,21,22,23]. It is also likely that ticagrelor-related dyspnea is caused by severe bronchospasms because adenosine causes bronchial smooth muscle cells to contract and stimulates the production of broncho-constrictive mediators [24,25,26,27,28].

The adenosine theory can also explain bradycardic episodes [29,30], since the interaction with adenosine receptors A1 and A2B causes activation of potassium channels at the level of the sinus and AV nodes, raising the potassium outward current while simultaneously inhibiting calcium-mediated slow channels, lowering the calcium inward current [31]. This causes suppression of the sinus node’s automaticity and increases the lead times of AV conduction, which ultimately results in bradycardia and conduction abnormalities.

Finally, by increasing the levels of adenosine, the drug activates the A2a and A2b receptors of leukocytes, exerting anti-inflammatory effects through inhibition of the release of pro-inflammatory cytokines such as IL6 and TNFα [32].

It has also been investigated why these side effects have an early onset following the acute event and tend to resolve with time. A possible explanation is that the initial coronary response to myocardial ischemia of any etiology is the release by the muscle cells of the ischemic myocardium of adenosine [33]. This explains why extracellular adenosine concentrations are markedly increased in the context of ACS treated with ticagrelor due to the ischemic “milieu”, cardiac release, and slowed cellular reuptake. As blood circulation is restored and the ischemic trigger resolves for reperfusion or myocardial tissue death, the adenosine levels tend to decline and, with them, the effects on lung smooth muscle and conduction tissue.

According to the adenosine theory, the decline in extracellular nucleoside levels justifies the gradual disappearance of the side effects over time [34].

Despite its apparent plausibility, there are many controversial aspects of the adenosine theory. According to certain investigations, there is no significant difference in plasma adenosine levels in individuals with and without ticagrelor-related dyspnea [35]. Likewise, there are no variations in the levels of circulating adenosine between ticagrelor patients and those receiving clopidogrel or prasugrel treatment [35,36,37].

#### 3.2.2. P2Y12 Receptors Hypothesis

P2Y12 receptors are found on various types of cells, including neuron cells, and microglia in the central nervous system. By blocking the activity of adenylyl cyclase and so lowering the levels of cAMP, activation of P2Y12 receptors reduces neuronal signaling. Blocking these receptors, ticagrelor may cause a gain of signaling of vagal C-fibers or glial cells which have the ability to activate the chemoreflex system in the brain and cause Cheyne–Stokes respiration [35,38].

Although this hypothesis is supported by less evidence, it is likely that the pathogenic mechanism of dyspnea is actually a combination of both hypotheses, with one mechanism that may prevail over the other depending on the case.

### 3.3. Therapy and Management

In most cases, modest, brief episodes of bradycardia and dyspnea happen during the first days of therapy [39,40]. It is advised to monitor the patient for some days while continuing the medication because the episodes will likely stop beyond that time [39,40,41,42]. Nonetheless, the patient may be forced to stop taking ticagrelor and to perform a switch of medication when these adverse effects become life-threatening or intolerably severe [43]. According to a published case study, theophylline infusion may have a favorable effect on certain symptoms, causing their remission [44]. Since this medication has never been used to treat patients in any clinical trials, neither the European Medicines Agency nor the Food and Drug Administration have made any recommendations in this regard yet.

About the necessity of implanting a permanent pacemaker in patients with high-grade AV block or symptomatic pauses, there are conflicting opinions. In the PLATO trial, the percentage of patients undergoing pacemaker implantation due to the presence of pathological pauses ascribed to ticagrelor was equal to 9% for pauses greater than 3 s and 9.4% for pauses greater than 5 s [11]. Several authors reported the need to implant a permanent dual chamber pacemaker in their patients because of persistent symptoms ten days after stopping ticagrelor [45]. Other authors, however, claimed that stopping ticagrelor alone caused second-degree AV block to revert to first-degree, suggesting that except in cases where the patient exhibits symptoms, routine use of a temporary or permanent pacemaker is not advised [46].

## 4. Conclusions

The reported clinical case is an example of how the knowledge of pharmacokinetics and pharmacodynamics and their possible relation to the side effects of antithrombotic therapy allows optimal post-revascularization clinical management. This approach may also provide the avoidance of further invasive procedures and, eventually, a reduction in healthcare costs.

In clinical practice, the occurrence of dyspnea, usually modest in entity, rarely requires ticagrelor interruption, and most patients may undergo clinical monitoring to ascertain the transitory nature of the adverse events. On the other side, some types of bradyarrhythmias may have more significant implications, such as the evaluation of the need for a pacemaker. In this setting, the ability to distinguish the pathogenesis and underlying cause of the adverse event would enhance subsequent clinical management. Indeed, although there is a pathogenetic difference between ticagrelor-induced bradyarrhythmias and those related to ischemia, the electrocardiographic pattern is often overlapping, and by itself, it does not provide definitive indications regarding the underlying etiological mechanism.

In the reported clinical case, the results of adenosine circulating levels supported the hypothesis of a side effect of ticagrelor related to elevated adenosine levels. It is worth noting that the patient on clopidogrel had experienced an ACS in the previous 24 h too, which could justify the slightly elevated values compared to the control patient since myocardial infarction itself can cause an increase in circulating adenosine levels released by cardiomyocytes following ischemic insult. Ascribing the occurrence of the events to a likely ticagrelor side effect led to the avoidance of pacemaker implantation, which could have been unnecessary. Nevertheless, these results are obviously only “hypothesis-generating” and we know that in the literature, there are contrasting data on the adenosine levels detectable in patients on ticagrelor.

Another debate is whether it is worthwhile to perform a switch between two antiplatelet drugs of similar efficacy (ticagrelor–prasugrel) or de-escalation of antiplatelet therapy to clopidogrel, supported by genetic and phenotypical tests, instead of continuing the drug confident that the side effect is transient. International guidelines do not give focused guidance, and there is no general consensus about this topic or direction toward the proposed strategies. Based on the results and evidence collected in this review, it is possible to suggest that neither a switch nor a downgrading of antiplatelet therapy is necessary except in rare cases where the side effects are clinically relevant and persistent over time. However, quantitative parameters based on which this clinical decision can be made have not yet been defined. In the reported clinical case, we applied the switch strategy with prasugrel in consideration of the symptomatic sinus-atrial block leading to syncope; however, we agree that for subclinical or less severe side effects, it is appropriate to continue and monitor ticagrelor therapy.

In conclusion, the clinical case offered: 1. the possibility of a literature revision on ticagrelor side effects and 2. a clinical model of real-world “precision medicine”.

## Figures and Tables

**Figure 1 ijms-24-10844-f001:**
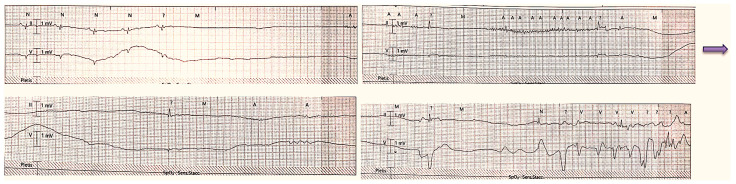
The electrocardiographic trace shows an atrial sinus block, resulting in a ventricular pause lasting 15 s. Following that, there is an additional ventricular pause of 4 s.

**Figure 2 ijms-24-10844-f002:**
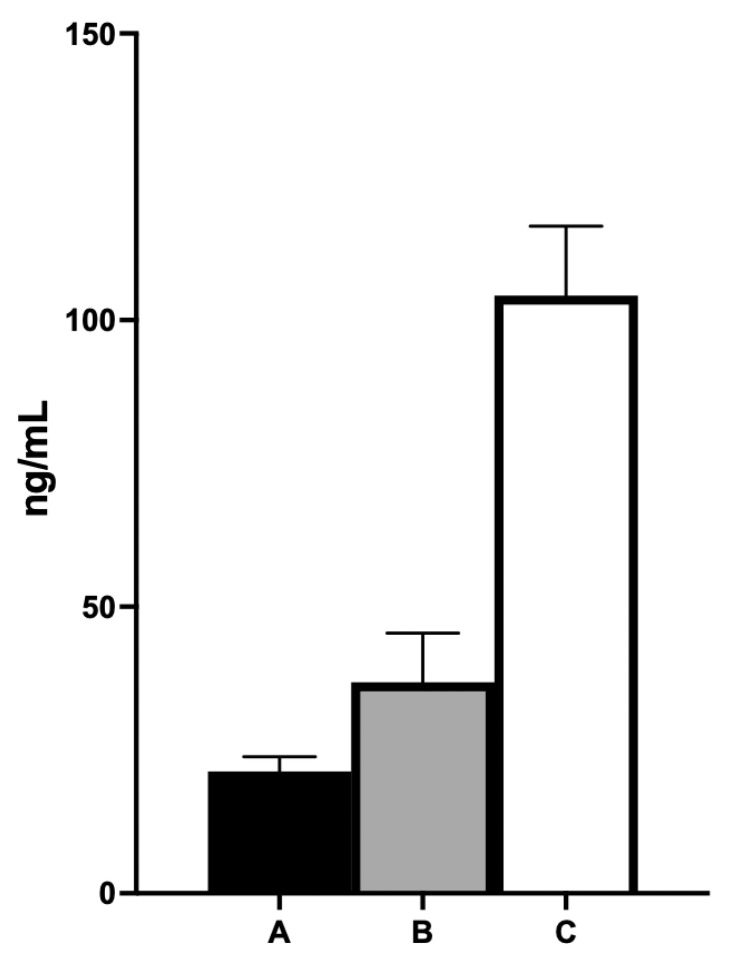
The graph presents a comparison of circulating adenosine levels among three subjects: a patient without antiplatelet therapy (**A**), a patient with recent acute coronary syndrome and on clopidogrel therapy (**B**), and a patient with recent acute coronary syndrome and on ticagrelor therapy (**C**). The values reported in each bar are obtained from a mean of four determinations for each subject.

## Data Availability

Not applicable.

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
