# Peer review of "Pathophysiological and Molecular Basis of the Side Effects of Ticagrelor: Lessons from a Case Report"

_ijms, 2023, doi:10.3390/ijms241310844_

Round 1

Reviewer 1 Report

This manuscript presents a ticagrelor case report (side effects) and then aims to gather the current available evidence regarding the molecular mechanisms underlying these side effects. They also propose possible decision-making algorithms regarding their management in clinical practice. This is a very interesting and important case-report and review of Ticagrelor side effects. The manuscript is very informative  and should be of interest to a broad readership in cardiovascular medicine. This referee has only minor comments / suggestions which may be helpful to finalize the review.

1)     The case presentation is nice, but could be shortened

2)     Could the authors give/cite the number of ticagrelor-treated patients in Europe or World-wide and then estimate incidence of side effects ? How many patients with such side effects can one estimate, i.e. for Italy?

3)     Adenosine levels (Fig2) I assume that each bar presents one patient (person) with (# = ???) determinations. There should be published data with adenosine levels before/during/after ticagrelor treatment. That information should be given.

4)     The authors have a nice and appropriate reference list.  Perhaps they can also extract further basic and clinical data on ticagrelor from the excellent review by Thomas MR/ Storey RF. (Thromb Haemost 2015;114-490-497)     

Well written.

Author Response

Point 1. The case presentation is nice but could be shortened.

Response 1. According to reviewer’s suggestion we shortened the case presentation.

Point 2.  Could the authors give/cite the number of ticagrelor-treated patients in Europe or World-wide and then estimate incidence of side effects? How many patients with such side effects can one estimate, i.e. for Italy? 

Response 2. We have addressed this issue and added the following sentences: “Considering the data provided by the distribution company in the European market [AstraZeneca S.p.A], there are currently approximately 650,000 patients receiving therapy with ticagrelor in Europe, of which around 61,000 in Italy. According to the percentages reported in the above mentioned randomized controlled trials (RCTs) about 8,500 patients may experience dyspnea and 3,700 different degrees of conduction disturbances in our Country.”

Point 3. Adenosine levels (Fig2) I assume that each bar presents one patient (person) with (# = ???) determinations. There should be published data with adenosine levels before/during/after ticagrelor treatment. That information should be given.

Response 3. As reported in the legend, each bar represents each patient. According to the reviewer’s suggestion, we specified that the values reported are a mean of 4 determinations for each subject. There are no solid data about adenosine levels in patients before or during ticagrelor treatment. However, it has already been described that adenosine concentration is higher in patients receiving ticagrelor, as compared with those receiving clopidogrel [see ref 20]. The purpose of our determinations was actually to confirm that adenosine may be involved in ticagrelor side effects, rather than providing a specific cut-off for treatment guidance.   

Point 4.  The authors have a nice and appropriate reference list.  Perhaps they can also extract further basic and clinical data on ticagrelor from the excellent review by Thomas MR/ Storey RF. (Thromb Haemost 2015;114-490-497)  

Response 4. We accepted the reviewer’s suggestion and added the reported reference (see ref. 32).

Reviewer 2 Report

The article entitled: „Pathophysiological and molecular basis of the side effects of ticagrelor: lesson from a case report” is an interesting case of a patient with ACS treated with ticagrelor who expiriences side efffects of the drug. This is a case presenting bradyarrhythmia, but there is nothing about ticagrelor related dyspnoe. It is worth to remember that ticagrelor has side effects like bradyarrhythmias and dyspnoe, however the case adds nothing to the current knowledge. The presented side effects were previosly reported in the DISPERSE trial and the PLATO trial. The possible pathophysiological explanations also have not changed or evolved since discovered. The article is overall well written and gathers all the neccessary information.

Good English quality, minor editing needed.

Author Response

As stated, the reported case report was about a significant bradyarrhythmic episode in a patient receiving ticagrelor after an ACS. Given the severity of such event, the subsequent management could have included a pacemaker implantation. However, we demonstrated that it probably would not be necessary if the event was considered as a ticagrelor side effect. In order to explain the pathogenesis of that bradyarrhythmic episode, we seize the opportunity to review the proposed hypotheses, focusing in particular on the role of adenosine increased levels. Since arrhythmias and dyspnea are actually the most common reported side effects of ticagrelor, and share the same pathophysiological explanation, we decided to mention both. The final purpose of our report was not to add any new hypotheses to the current knowledge, but rather we aimed at proposing a reasoned management of those side effects.